# Overexpression of the *Glycyrrhiza uralensis* Phenylalanine Ammonia-Lyase Gene *GuPAL1* Promotes Flavonoid Accumulation in *Arabidopsis thaliana*

**DOI:** 10.3390/ijms26094073

**Published:** 2025-04-25

**Authors:** Xifeng Chen, Chao Jiang, Mengqian Long, Xiangxiang Hu, Shouhao Xu, Haotong Huo, Ruixin Shi, Qing Xu, Shuangquan Xie, Zihan Li, Haitao Shen, Fei Wang, Guanghui Xiao, Quanliang Xie, Shandang Shi, Hongbin Li

**Affiliations:** 1Key Laboratory of Xinjiang Phytomedicine Resource and Utilization of Ministry of Education, Key Laboratory of Oasis Town and Mountain-Basin System Ecology of Bingtuan, College of Life Sciences, Shihezi University, Shihezi 832000, China; cxf_cc@shzu.edu.cn (X.C.); 20232006024@stu.shzu.edu.cn (C.J.); 20232006044@stu.shzu.edu.cn (M.L.); 18890626379@163.com (X.H.); 18171481781@163.com (S.X.); 20221006012@stu.shzu.edu.cn (H.H.); 13779851963@163.com (R.S.); xq19901526963@163.com (Q.X.); xiesq@shzu.edu.cn (S.X.); shtshz-bio@shzu.edu.cn (H.S.); feiw@shzu.edu.cn (F.W.); xiequanliang001@shzu.edu.cn (Q.X.); 2Department of Civil, Environmental, and Construction Engineering, College of Engineering and Computer Science, University of Central Florida, Orlando, FL 32816, USA; zihl2721@gmail.com; 3National Key Laboratory of Cotton Bio-Breeding and Integrated Utilization, School of Life Science, Henan University, Kaifeng 475000, China; guanghuix@snnu.edu.cn

**Keywords:** phenylalanine ammonia-lyase, *Glycyrrhiza* species, flavonoid biosynthesis, transcriptomic analysis, *Arabidopsis*

## Abstract

Phenylalanine ammonia-lyase (PAL) serves as a pivotal regulatory enzyme at the initial branching point of the phenylpropanoid pathway, exerting a profound influence on downstream reactions essential for flavonoid biosynthesis. *Glycyrrhiza* species are important medicinal plants and provide plenty of roots as raw materials for further utilization, with the components of glycyrrhizic acid and flavonoids as two major active ingredients. However, functional studies of the *PAL* genes in the medicinal *Glycyrrhiza* species remain limited. In this study, we identified seven *PAL* family genes from each of the three medicinal *Glycyrrhiza* species, *Glycyrrhiza uralensis* Fisch., *G. inflata* Bat., and *G. glabra* L., and comprehensively analyzed their phylogenetic relationships, gene structures, motif distributions, and promoter *cis*-elements. Gene expression profiling revealed that *PAL1* is highly expressed in roots and significantly induced by drought and salt stresses. We further selected *G. uralensis GuPAL1* for functional investigation in *Arabidopsis*. *GuPAL1*-overexpression lines (*GuPAL1*-OE) demonstrated significant enhancements in plant growth, flavonoid accumulation, and hormone levels in *Arabidopsis thaliana*. Conversely, the *Atpal1* mutant plants displayed marked reductions in these traits, while the transgenic lines of *GuPAL1-*OE in the *Atpal1* mutant (*Atpal1*/*GuPAL1*) recovered to the normal phenotypes similar to wild type (WT). Transcriptomic analysis of the *GuPAL1*-OE plants compared to WT demonstrated that several key genes in the phenylpropanoid and flavonoid metabolic pathways (*4CL*, *CCoAOMT*, *CAD*, *POD*, *F3H*, *FLS*) were significantly enriched, suggesting that *GuPAL1* may promote plant growth and flavonoid biosynthesis by regulating diverse cellular functions, metabolic pathways, and associated gene expressions. These findings highlight the functional importance of *GuPAL1* in flavonoid biosynthesis, and provide valuable insights into the molecular mechanisms underlying the medicinal properties of *Glycyrrhiza* species.

## 1. Introduction

Licorice is a widely used plant that plays an important role in various fields, including medicine, food, cosmetics, animal husbandry, and ecology. In the medicinal field, the Chinese Pharmacopoeia has officially recorded the dried roots and rhizomes of the leguminous plants *G. uralensis*, *G. inflata*, and *G. glabra* as medicinal licorice since the 2005 edition [1]. In the food industry, licorice extract is widely used as a natural sweetener in products such as chewing gum, candy, and alcoholic beverages, aligning with the development trend of health foods [2]. In the cosmetics industry, licorice root extract offers a safe and effective option for treating skin issues such as melasma and hyperpigmentation [3]. In animal husbandry, licorice, as a feed additive, can improve poultry growth performance, feed efficiency, antioxidant levels, and immunity, and has positive effects on the immune system, liver, and lung diseases [4]. In the ecological field, licorice can be used to ameliorate salinized soils, as its rhizobium can fix nitrogen and enhance soil fertility. Co-inoculation of licorice with rhizobia and Pseudomonas significantly increases plant biomass, nodulation, and nitrogen content [5]. Additionally, licorice is abundant in a variety of bioactive compounds, particularly flavonoids, which exhibit significant pharmacological properties, including antioxidant, anti-inflammatory, antimicrobial, and anticancer effects. These bioactive flavonoids have not only underscored the medicinal value of licorice but also propelled the development of novel licorice-derived products [6]. With the escalating market demand for licorice, wild resources have experienced a dramatic decline, rendering artificial cultivation a critical strategy to address the resource shortage. The cultivation of licorice, especially the three *Glycyrrhiza* species of *G. uralensis*, *G. inflata*, and *G. glabra*, particularly for its flavonoid-rich extracts, is essential to meet both industrial and therapeutic needs.

The Chinese Pharmacopoeia and other regulations specify the content range of glycyrrhizin and liquiritin in licorice [1], but the medicinal component content in artificially cultivated licorice is relatively low, especially liquiritin, which often fails to meet the standards. Liquiritin is a kind of flavonoid, and *G. uralensis* primarily grows in desert or saline-alkaline soils [7]. Environmental stress, such as salt and drought, can promote the accumulation of flavonoids in *G. uralensis*, enhancing both their content and quality [8]. However, the molecular mechanisms underlying this phenomenon remain unclear. This study aims to elucidate the molecular mechanisms underlying flavonoid accumulation under salt and drought stress in *G. uralensis*. The findings will provide both theoretical foundations for stress response understanding and practical insights for cultivation enhancement.

Flavonoids are an important class of polyphenolic compounds in plant secondary metabolites. Their synthesis is influenced by various factors, including light [9,10,11,12], temperature [13,14,15,16,17], herbivorous insects [18], pathogens [19,20], plant pollination [21], C/N ratio [22,23], and plant hormones [24,25,26,27,28,29,30,31,32,33,34,35,36,37,38,39,40,41,42]. The biosynthesis of flavonoids begins with the shikimate pathway, where carbohydrates are converted into the aromatic amino acid phenylalanine. This then enters the flavonoid biosynthesis pathway through the phenylpropanoid pathway, which is a branch of phenylpropanoid metabolism [43]. The phenylalanine pathway is crucial in plant metabolism as it facilitates the transition from primary to secondary metabolism. PAL is the key rate-limiting enzyme in this pathway, catalyzing the conversion of phenylalanine to cinnamic acid [44,45]. PAL is positioned at the upstream node of the metabolic network, directly determining substrate availability and influencing all downstream reactions of flavonoid biosynthesis. Its activity and expression not only regulate the types and yields of flavonoids but also impact the plant’s physiological and biochemical characteristics and environmental adaptability [46,47].

In the 1960s, Koukol et al. successfully isolated the first *PAL* gene from plants [48]. Since then, *PAL* genes have been successfully isolated from a variety of plants, including *Solanum tuberosum*, *soybean*, *Brassica juncea*, *ginkgo*, and *Brassica napus* L. [49,50,51,52]. The *PAL* gene family is extensive, with significant variation in the number of members between different species, though changes at the coding sequence level are relatively small [53,54]. Currently, there are few reports on the *PAL* gene family and its functions in medicinal *Glycyrrhiza* species. In this study, we identified seven *PAL* genes from three *Glycyrrhiza* species, characterizing their phylogeny, structure, and regulatory elements. *GuPAL1* showed significant upregulation under salt/drought stress. In *Arabidopsis*, *GuPAL1* overexpression enhanced stress tolerance compared to the WT and *Atpal1* mutant. Transcriptomics revealed *GuPAL1*’s role in activating flavonoid biosynthesis through metabolic and transcriptional reprogramming, demonstrating its central regulatory function.

## 2. Results

### 2.1. Identification of the PAL Gene Family in Three Medicinal Glycyrrhiza Species

We performed a BLAST (Version 2.14.0) search using the *A. thaliana PAL* gene sequences against the genomes of three medicinal *Glycyrrhiza* species to identify the *PAL* genes present in each species. The identified *PAL* genes were then submitted to the Pfam, CDD, and SMART databases for validation of the presence of the PAL domain. Through this process, we identified seven members of the *PAL* gene family in each *Glycyrrhiza* species. Based on the chromosomal positions of each *PAL* gene and the phylogenetic relationships among the *PAL* genes from the three *Glycyrrhiza* species, we assigned names to the genes and compiled a correspondence between gene ID numbers and gene names, as shown in Appendix A. The coding sequences of the *PAL* genes analyzed exhibited considerable variation in length, ranging from 374 amino acids (AA) (GgPAL6) to 737 AA (GgPAL1, GiPAL1, GuPAL1), with an average of approximately 690 AA per protein. The molecular weights (MW) of the PAL proteins varied between 40,463.41 Da (GgPAL6) and 78,776.78 Da (GgPAL4). Theoretical isoelectric point (pI) values were predominantly acidic (pI < 7), except for GgPAL6 (pI 8.51) and GuPAL4 (pI 8.55), which exhibited basic pI values. Furthermore, all PAL proteins displayed negative Grand Average of Hydropathicity (GRAVY) values (−0.181 to −0.006), confirming their hydrophilic nature (Appendix A).

### 2.2. Chromosomal Distribution Patterns of PAL Gene Family in Three Medicinal Glycyrrhiza Species

Through the analysis of the distribution of *PAL* genes on the chromosomes of *G. glabra*, *G. inflata*, and *G. uralensis*, it was observed that the distribution of *PAL* genes on the chromosomes is highly consistent across the three *Glycyrrhiza* species (Figure 1). Among them, the *PAL3*–*PAL5* genes are clustered on chromosome 2. These closely positioned genes are likely functionally related, potentially co-participating in shared biological processes through similar transcriptional regulation mechanisms or interactions. In contrast, other *PAL* genes exhibit a dispersed distribution across the genome. This dispersed arrangement enhances gene expression flexibility and functional diversity, enabling organisms to adapt more effectively to environmental changes. Such a distribution pattern is the result of long-term evolutionary selection, facilitating the coordination of diverse biological functions and meeting the demands of survival and reproduction.

### 2.3. Phylogenetic Analysis of the PAL Gene Family in Three Medicinal Glycyrrhiza Species, Soybean, and A. thaliana

To enhance the classification and evolutionary relationships of *PALs*, a phylogenetic tree was constructed using the PAL protein sequences from *G. uralensis*, *G. inflata*, *G. glabra*, *Soybean*, and *A. thaliana*. The 33 proteins were clustered into three branches, Clade A, Clade B, and Clade C, in the phylogenetic tree (Figure 2). The Clade A branch includes *PAL* gene members from *A. thaliana* (*AtPAL1*–*AtPAL4*), soybean (*GmPAL5*, *GmPAL8*), and the three *Glycyrrhiza* species (*GuPAL1*, *GiPAL1*, *GgPAL1*). The Clade B branch also gathers *PAL* genes from multiple species, including soybean’s *GmPAL1*, and genes from *Glycyrrhiza (GuPAL6*, *GuPAL7*, *GiPAL6*, *GiPAL7*, *GgPAL6*, *GgPAL7).* The Clade C branch contains soybean’s *GmPAL2*, *GmPAL3*, *GmPAL4*, *GmPAL6*, and *GmPAL7* genes, as well as genes from *Glycyrrhiza (GuPAL2*–*GuPAL5*, *GiPAL2*–*GiPAL5*, *GgPAL2*–*GgPAL5*). The clustering patterns of *PAL* gene members across species reflect both the conservation and specificity of their family evolution. This provides clues for exploring the evolution of gene functions and offers molecular evidence for understanding the evolution of plant secondary metabolism.

### 2.4. Analysis of Collinearity Relationships Among PAL Gene Families on Chromosomes in Three Medicinal Glycyrrhiza Species

To explore the potential evolutionary mechanisms of the *PAL* gene family in the three medicinal *Glycyrrhiza* species, an analysis of collinearity relationships was conducted on the genomes of *G. uralensis*, *G. inflata*, and *G. glabra* (Figure 3). From the figure, it is apparent that there are extensive collinear relationships among the chromosomes of the three *Glycyrrhiza* species, suggesting that their genomic structures have been largely preserved during evolution. The discovery of these collinear relationships provides important evidence for in-depth studies on the evolutionary relationships of *Glycyrrhiza* species.

### 2.5. Analysis of Promoter Cis-Acting Elements, Conserved Motifs, and Gene Structures of PAL Gene Family

We conducted an in-depth analysis of the structural and functional diversity of the *PAL* gene family from multiple perspectives, including element distribution, evolutionary relationships, motif distribution, and CDS structure (Figure 4). The distribution of *cis*-elements varies among different genes, with only closely related genes exhibiting similar distribution patterns. This may be associated with the unique expression patterns of genes in different tissues, developmental stages, or environmental conditions (Figure 4A). The heatmap of *cis*-element counts reveals that the most abundant elements are predominantly concentrated on *PAL1*, including G-box, GT1-box, and ABRE, all of which are stress-related, suggesting that *PAL1* is likely induced by stress. This is followed by *PAL2* (Figure 4B). The results of domain and gene structure analyses indicate that most genes share conserved domains, and closely related genes exhibit similar gene structures. Notably, *PAL1* and *PAL2* contain more exons than other genes, reflecting their potential functional complexity and evolutionary adaptability. This structural diversity may contribute to their specialized roles in stress responses and metabolic regulation. These findings suggest that *PAL1* likely plays a critical role in the stress response of *Glycyrrhiza* plants.

### 2.6. Analysis of Tissue-Specific and Stress-Induced Expression of PAL Gene Family in Three Medicinal Glycyrrhiza Species

By revealing the expression patterns of *PAL* genes in different tissues and under stress conditions in the three medicinal *Glycyrrhiza* species, we have provided important insights into their roles in plant growth, development, and stress response (Figure 5). The tissue expression heatmap reveals that *PAL1* and *PAL2* are ubiquitously expressed across all tissues, with *PAL1* exhibiting particularly high expression levels in the roots and stems of *G. uralensis* (Gura) as well as in the roots of *G. inflata* (Ginf) and *G. glabra* (Ggla) (Figure 5A). This suggests that *PAL1* is closely associated with root metabolite content. Under salt and drought stress treatments, *PAL1* and *PAL2* showed the highest expression levels, with *PAL1* being significantly induced, consistent with the earlier analysis of promoter *cis*-elements (Figure 4 and Figure 5B). These findings indicate that PAL1, particularly *GuPAL1*, is a key stress-responsive gene and strongly linked to root metabolite accumulation, making it a promising candidate for further investigation.

### 2.7. Phenotypic Analysis of Wild-Type, GuPAL1-OE, Atpal1 Mutant, and Atpal1/GuPAL1 A. thaliana Plants

By transforming WT and *Atapl1* mutant (*SALK_000357*) *A. thaliana* plants with *35S::GuPAL1* and obtaining T3 generation plants, the study analyzed the role of *GuPAL1* in promoting *A. thaliana* growth and flavonoid biosynthesis. The WT, *GuPAL1*-OE, *Atpal1* mutant, and *Atpal1*/*GuPAL1 A. thaliana* seedlings were transplanted into nutrient soil and their phenotypes were continuously observed and recorded. As shown in Figure 6A, compared to the WT 15 days after transplantation, the *GuPAL1*-OE lines exhibited more vigorous growth, with a significant increase in rosette leaf number and a noticeable enlargement of leaves. However, the *Atpal1* mutant lines showed a decrease in rosette leaf number and smaller leaf size. The *Atpal1*/*GuPAL1 A. thaliana* plants displayed a growth phenotype similar to the WT. Further comparisons of plant growth at 25 and 45 days were conducted. At 25 days, the *GuPAL1*-OE showed significantly greater plant height than both the WT and *Atpal1* mutant, while the *Atpal1*/*GuPAL1* functional complement plants exhibited heights similar to the WT (Figure 6B). By 45 days, the *GuPAL1*-OE plants maintained a clear growth advantage, with significantly longer overall plant length compared to the WT and *Atpal1* mutant, whereas the *Atpal1* mutant functional complement plants displayed lengths similar to the WT (Figure 6C). These results suggest that overexpression of *GuPAL1* significantly promotes *A. thaliana* growth, while the growth of the *Atpal1* mutant is inhibited, further confirming the key role of *GuPAL1* in plant growth and development.

To further analyze the impact of *GuPAL1* on plant metabolism, the flavonoid content in the underground parts and whole plants of WT, *GuPAL1*-OE, *Atpal1* mutant, and *Atpal1*/*GuPAL1 A. thaliana* was measured. Compared to the WT, the *GuPAL1*-OE *A. thaliana* had a significantly higher flavonoid content, while the *Atpal1* mutant exhibited a decrease in flavonoid content. The flavonoid content in the *Atpal1*/*GuPAL1* plants showed no significant difference from the WT, indicating that *GuPAL1* may be involved in the flavonoid biosynthesis pathway (Figure 6D,E).

### 2.8. Phenotypic Analysis of WT, GuPAL1-OE, Atpal1 Mutant, and Atpal1/GuPAL1 A. thaliana Plants

To elucidate the potential functions of *GuPAL1* in cellular physiological states, metabolic pathways, and gene expression regulation, Gene Ontology (GO) and Kyoto Encyclopedia of Genes and Genomes (KEGG) analyses were performed on up-regulated genes identified in the transcriptome of *GuPAL1*-OE *A. thaliana*. GO analysis demonstrated that the genes were significantly enriched in categories such as cellular processes, response to stimuli, biological regulation, developmental processes, and metabolic activities, suggesting that *GuPAL1* may affect *A. thaliana* growth, development, and stress responses by regulating various cellular functions and biological processes (Figure 7A). As shown in Figure 7B, the KEGG analysis revealed that pathways with a large number of genes included metabolism, genetic information processing, and signal transduction. Further analysis indicated significant enrichment in pathways such as plant hormone signal transduction, flavonoid biosynthesis, phenylpropanoid biosynthesis, and starch and sucrose metabolism (Figure 7C). *GuPAL1* may regulate hormone synthesis, thereby affecting plant hormone signal transduction pathways. These findings suggest that *GuPAL1* may influence *A. thaliana* metabolism, particularly flavonoid biosynthesis-related pathways, through the regulation of plant hormone signaling and metabolic processes.

In summary, the up-regulated genes in *GuPAL1*-OE *A. thaliana* are significantly enriched in multiple GO and KEGG categories, and may promote the synthesis and accumulation of flavonoids by regulating cellular physiological states, metabolic pathways, and gene expression. These findings provide important clues for elucidating the regulatory mechanisms of *GuPAL1* in flavonoid biosynthesis.

### 2.9. Expression Pattern of Up-Regulated Genes in the Transcriptome of GuPAL1-OE A. thaliana

By screening the up-regulated genes in the transcriptome of *GuPAL1*-OE *A. thaliana*, it was found that the expression of several key genes in the phenylpropanoid and flavonoid metabolic pathways was significantly up-regulated (Figure 8A). In the phenylpropanoid metabolic pathway, the up-regulated genes included those encoding *4*-*coumarate*–*CoA ligase (4CL)*, *cinnamate*–*4*-*hydroxylase (CAD)*, *caffeoyl*–*CoA O*–*methyltransferase (CCoAOMT)*, and *peroxidase (POD).* In the flavonoid metabolic pathway, the up-regulated genes included those encoding *chalcone isomerase (CHI)*, *flavanone 3*-*hydroxylase (F3H)*, *flavonol synthase (FLS)*, *cytochrome P450 monooxygenase 75B1 (CYP75B1)*, and *anthocyanidin reductase (ANR).* This coordinated upregulation suggests a metabolic flux redirection toward flavonoid production, with *CHI* and *F3H* enhancing the core flavonoid biosynthesis, *FLS* and *CYP75B1* promoting flavonol diversification, and *ANR* facilitating the conversion to proanthocyanidins, collectively contributing to the observed increase in total flavonoid content in *GuPAL1*-OE plants. Additionally, in the hormone signal transduction pathways, genes up-regulated in the cytokinin signaling pathway included *isopentenyltransferase (IPT)*, *UDP*–*glycosyltransferases (UGT73C1*, *UGT76C1)*, and *cytokinin oxidase (CKX)* (Figure 8B). Up-regulated genes in the *salicylic acid (SA)* pathway included pathogenesis-related gene *non*-*expressor 1 (NPR1)* and *pathogenesis*-*related protein 1 (PR*-*1)* (Figure 8C). In the abscisic acid (ABA) pathway, the up-regulated genes included ABA receptors (*PYR*/*PYL*), *protein phosphatase 2C (PP2C)*, *sucrose non*-*fermenting protein kinase 2 (SnPK2)*, and ABA-responsive element binding factors (*ABF*) (Figure 8D).

To verify the impact of *GuPAL1* on these metabolic pathways, the levels of zeatin (ZT), SA, and ABA were measured in WT, *GuPAL1*-OE, *Atpal1* mutant, and *Atpal1*/*GuPAL1 A. thaliana* plants. The results showed that the *GuPAL1*-OE plants had significantly elevated levels of ZT, SA, and ABA in both the whole plant, underground parts, and aerial parts, while the *Atpal1* mutant exhibited significantly lower levels of these hormones, and the hormone levels in the complement plants were similar to the WT (Figure 8E–G). These findings suggest that *GuPAL1* not only enhances flavonoid accumulation by promoting the phenylpropanoid and flavonoid metabolic pathways but also significantly increases the synthesis of plant hormones such as ZT, SA, and ABA. These findings position *GuPAL1* as a prime molecular target for genetically improving *Glycyrrhiza* varieties, enabling simultaneous enhancement of medicinal flavonoid production and stress adaptability in cultivation practices.

### 2.10. Protein–Protein Interaction (PPI) Prediction Analysis of GuPAL1

Based on the analysis results of the *GuPAL1*-OE transcriptome data, protein–protein interaction (PPI) prediction analysis was conducted using the *GuPAL1* gene (AT2G37040) and genes from the four aforementioned pathways. The analysis revealed that *GuPAL1* interacts with 10 flavonoid pathway genes including *AT1G24735 (CCoAOMT)*, *AT1G61720 (ANR)*, *AT3G21230 (4CL)*, *AT3G21240 (4CL)*, *AT3G51240 (F3H)*, *AT4G26220 (CCoAOMT)*, *AT5G05270 (CHI)*, *AT5G07990 (CYP75B1)*, *AT5G08640 (FLS)*, *AT5G63590 (FLS)*, and 2 genes from the SA synthesis pathway of *AT5G45110 (NPR1*) and *AT2G14610 (PR1)*, which are co-expressed. Additionally, 38 other genes were also found to co-express with *GuPAL1* (Figure 9). These interacting genes provide crucial targets for subsequent metabolomic analyses, enabling in-depth elucidation of flavonoid biosynthetic mechanisms through approaches like CRISPR-based validation of key regulatory nodes and construction of regulatory networks.

## 3. Discussion

In recent years, increasing evidence has shown that *PAL* genes play a crucial role in plant growth, development, and resistance [55]. Although the functions of the *PAL* gene family have been extensively studied in various plant species through high-throughput sequencing [56,57], research on *Glycyrrhiza* is still limited. Therefore, this study systematically explored the functions of the *PAL* gene family in three medicinal *Glycyrrhiza* species *(G. uralensis*, *G. inflata*, and *G. glabra)*, with a focus on revealing the molecular mechanisms of *GuPAL1* in flavonoid biosynthesis and plant growth.

*PAL* genes have been reported in several species, including *Glycine max*, *A. thaliana*, *Solanum tuberosum*, and *Medicago sativa* [58,59], with varying numbers of family members in different species. Seven *PAL* genes identified in the three *Glycyrrhiza* species, along with their chromosomal collinearity relationships, reveal the evolutionary conservation of the *Glycyrrhiza PAL* gene family. Phylogenetic analysis categorized the *PAL* genes into three clades (A, B, and C). *GuPAL1* clustered with *A. thaliana AtPAL1*–*AtPAL4* and soybean *GmPALs* [60,61,62] in Clade A, indicating its conserved function in phenylpropanoid metabolism.

Previous studies have found that disruption of the *PAL1* and *PAL2* genes in *A. thaliana* makes mutant plants particularly sensitive to environmental growth conditions, affecting their fertility [54]. The disruption of the other two *PAL* genes altered the expression of genes involved in phenylpropanoid biosynthesis, as well as carbohydrate and amino acid metabolism [61]. There was an overaccumulation of Phenylalanine, while the three major flavonol glycosides and lignin monomers were significantly reduced [52,61]. Promoter element analysis revealed a significant enrichment of stress-related binding sites on the *PAL1* gene. Tissue-specific expression analysis demonstrated that *PAL1* and *PAL2* are highly expressed across all tissues of the three medicinal *Glycyrrhiza* species, with the highest expression levels observed in the root and stem tissues. Under salt and drought stress conditions, the expression of *PAL1* and *PAL2* was significantly up-regulated, with *PAL1* exhibiting the most pronounced increase. These findings suggest that *PAL1* and *PAL2* may play a crucial role in the plant’s response to environmental stress, particularly *PAL1*, whose expression pattern is closely associated with flavonoid biosynthesis. These results provide important insights for further investigation into the role of *PAL1* in flavonoid synthesis.

Through overexpression of *GuPAL1* in *A. thaliana* lines, this study confirmed the important role of *GuPAL1* in promoting plant growth and flavonoid synthesis. Compared to the WT, the *A. thaliana* plants overexpressing *GuPAL1* exhibited an increased number of rosette leaves and larger leaf size 15 days after transplantation. By day 25 and day 45, the plant height of these overexpressing lines was significantly greater than that of the WT and the *Atpal1* mutant. Moreover, flavonoid content analysis revealed that the flavonoid levels in the underground parts and the whole plants of *GuPAL1*-OE *A. thaliana* were significantly higher than those of the WT, while the *Atpal1* mutant showed a reduced flavonoid content. The flavonoid content in the *GuPAL1*-transformed mutant did not significantly differ from that of the WT. These results suggest that *GuPAL1* plays a crucial role in flavonoid synthesis.

Previous studies have shown that 4CL contributes to channeling the flux of various phenylpropanoid biosynthetic pathways [63]. Rui Wu et al. discovered in their study of Prunus mume that FLS is highly expressed, and a higher number of flavonoid compounds, including flavones, were detected. Enzymes associated with the flavonoid pathway, such as F3H, FLS, and ANS, all belong to the 2-oxoglutarate-dependent dioxygenase (2-ODD) subfamily [64,65]. F3H is a key enzyme that directs carbon flux toward the biosynthesis of 3-hydroxy flavonoid compounds, responsible for the biosynthesis of flavonols and anthocyanins [66]. Transcriptomic analysis revealed significant enrichment in pathways related to plant hormone signal transduction, flavonoid biosynthesis, and phenylpropanoid biosynthesis. Furthermore, expression pattern analysis demonstrated that key enzyme genes involved in phenylpropanoid and flavonoid metabolic pathways, as well as hormone signaling pathways, were significantly up-regulated. Protein–protein interaction predictions indicated co-expression between *GuPAL1* and flavonoid biosynthetic enzymes (4CL, FLS, F3H, etc.) and SA pathway proteins (NPR1, PR-1), suggesting that *GuPAL1* may enhance pathway efficiency by stabilizing metabolic complexes or facilitating substrate channel coordination, which is consistent with previous studies [67].

## 4. Materials and Methods

### 4.1. Plant Materials

Using the PAL family protein sequences from *A. thaliana* and *G. max* as references, the *PAL* family members in the genomes of *G. uralensis*, *G. inflata*, and *G. glabra* were identified through local BLASTP searches. The protein kinase domain (PLN02457) was validated using the InterProScan server (http://www.ebi.ac.uk/interpro/, accessed on 10 June 2024) and manually reviewed. The physical locations of the *PAL* family members on the chromosomes were mapped using TBtools software (Version 2.056) [68]. We compared protein sequences using MEGA 11 software (Version 11.0.13) and constructed a Neighbor-Joining (NJ) tree with 1000 bootstrap replicates [69], and the phylogenetic tree was visualized using the online tool Evolview (https://evolgenius.info/, accessed on 10 June 2024) [70]. Conserved domains were analyzed using BioEdit (Version 7.0.9) and MEME (https://meme-suite.org/meme/, accessed on 10 June 2024) [71] online software. These sequences were submitted to MCScanX (Multiple Collinearity Scan) and Dual Synteny Plotter tools to analyze the syntenic relationships between the three *Glycyrrhiza* species [72]. Based on *Glycyrrhiza* genomic data and annotation files, TBtools software (Version 2.056) was used to obtain the 2000 bp DNA sequences upstream of the *PAL* family genes. The PlantCARE website (https://bioinformatics.psb.ugent.be/webtools/plantcare/html/, accessed on 10 June 2024) was used to predict cis-regulatory elements in the *PAL* gene promoter regions, and TBtools (Version 2.056) was used for visualization. Using the PAL family protein sequences from the three *Glycyrrhiza* species, *A. thaliana*, and *Glycine max*, the MEME (https://meme-suite.org/meme/, accessed on 10 June 2024) was employed to analyze conserved protein motifs, with each motif having a *p*-value lower than 1 × 10^−5^. The XML files generated by the MEME program, the phylogenetic tree’s NWK files, and the GFF files for gene structure were obtained. Then, using TBtools (Version 2.056), the gene structure, phylogenetic tree, and conserved motifs were predicted and visualized.

### 4.2. Tissue-Specific Expression Materials and Stress Treatment in Three Medicinal Glycyrrhiza Species

Due to the low germination rate of wild *Glycyrrhiza* seeds, this study employed a 35 min treatment with 98% sulfuric acid to break seed dormancy, followed by sterilization with 0.1% mercuric chloride [73]. The sterilized seeds were then placed in a controlled artificial climate chamber under the following conditions: 16 h of light per day, humidity between 50 and 55%, and day/night temperatures of 28 °C and 25 °C, respectively, using vermiculite as the substrate and irrigating with sodium-free 1 × Hoagland nutrient solution. After 45 days of cultivation, root, stem, and leaf tissues were collected for tissue-specific expression analysis. For salt stress treatment, 42-day-old seedlings were first hydroponically cultured in sodium-free 1 × Hoagland nutrient solution for 3 days, then treated with a nutrient solution containing 200 mM NaCl. Three biological replicates were set, with root materials harvested at 2 h and 24 h after treatment and stored at −80 °C for later analysis. The method for drought stress treatment was similar to that for salt stress. After the same pretreatment, seedlings were treated with a nutrient solution containing 10% PEG6000. Again, three biological replicates were set, with root materials harvested at 2 h and 24 h after treatment and stored at −80 °C for further analysis.

### 4.3. Expression Pattern of PAL Family Members in Three Medicinal Glycyrrhiza Species

qRT-PCR primers were designed using Primer 5.0 (Appendix A), and total RNA was extracted from three biological replicates using the RNAprep pure plant kit. After determining RNA concentration with a spectrophotometer and verifying integrity via agarose gel electrophoresis, 1 μg of RNA was reverse-transcribed into cDNA with three technical replicates. *Glycyrrhiza* actin gene *GuActin* was used as an internal reference (Appendix A). qRT-PCR was performed using the TianGen reagent on the Roche480 system with 40 cycles of amplification. Data were processed using the 2^−ΔΔCt^ method, and heatmaps were generated using R (Version 4.3.2) software.

### 4.4. Construction of Plant Expression Vectors and Transformation of A. thaliana

Using reverse-transcribed cDNA from *G. uralensis* as a template, the target fragment was amplified by PCR using the following primers: upstream primer 5′ATGATGGAGTTTTCCAATG3′ and downstream primer 5′CTAGCATATGGGAGAG3′. The PCR product was analyzed by 1% agarose gel electrophoresis, yielding a specific band of 2181 bp (Appendix A). Subsequent sequencing and alignment confirmed that this band corresponds to the *GuPAL1* gene fragment. The *GuPAL1* gene was then cloned into the *pMD19*-*T* vector. Both the recombinant plasmid *pMD19*-*T*-*PAL1* and the expression vector *pCAMBIA2300* were digested with the restriction enzymes BamHI and XbaI, and the target fragment was recovered and ligated. After plasmid extraction, the recombinant plasmids were verified by BamHI and XbaI digestion and sequencing analysis, resulting in the recombinant plasmid *35S::GuPAL1*. WT *A. thaliana* seeds were sterilized and vernalized, then cultured and transplanted into soil. During the flowering stage, Agrobacterium containing the recombinant plasmid was suspended and infiltrated into the flower buds. After dark incubation for 24 h, the plants were returned to normal growth conditions, and the infiltration was repeated three times at 5–7-day intervals to collect T0 seeds. The T0 seeds were sown on MS medium containing kanamycin, and genomic DNA was extracted from 3-week-old seedlings for PCR identification. Positive plants were selected for T1 seed collection. The DNA-positive T1 seeds were sown on kanamycin-containing medium, and 3-week-old seedlings were used for RNA extraction. After reverse transcription, qRT-PCR was performed using *GuActin* as an internal reference (Appendix A), and high-expressing lines were selected for T2 seed collection (Appendix A).

### 4.5. Quantification of Flavonoids and Hormones in GuPAL1-OE A. thaliana Using LC–MS/MS

T2 seeds from three independent transgenic lines were sown on MS medium (triplicate plates per line), and after transplanting to nutrient soil for 45 days, whole plants and roots of T3 *Arabidopsis* were collected from each biological replicate. The tissues were ground into powder, and methanol was added for extraction. The samples underwent ultrasonic treatment, centrifugation, and the supernatant was evaporated under nitrogen. After reconstitution and further centrifugation, the final samples were prepared for analysis. A series of calibration standard solutions was prepared (including IAA, ZT, BR, SA, ABA, MeJA, GA3, and Rutin), and samples were injected with different concentrations of standards. Standard curves were created based on peak areas and concentrations, with good linearity in specific concentration ranges (Appendix A). The analysis was performed using electrospray ionization (ESI) as the ion source, with multiple reaction monitoring (MRM) mode. The desolvation temperature, ion source temperature, gas flow rate, and capillary voltage were optimized, and the appropriate parent and daughter ions for quantitative analysis were identified (Appendix A). Chromatographic separation was conducted using a Waters ACQUITY UPLC BEH C18 column under the specified flow rate, injection volume, and column temperature, with a gradient elution program (Appendix A).

### 4.6. Transcriptome Analysis

The raw sequencing data from three biological replicates were subjected to quality control and filtering using FastQC (Version 0.12.0) and Trimmomatic (Version 0.39) to remove low-quality data. The cleaned data were mapped to the *A. thaliana* reference genome, and the files were converted into the appropriate format. Transcript data and expression matrices (count values) were obtained. Gene expression levels were calculated using the StringTie (Version 1.3.3b) software and quantified as Fragments Per Kilobase of transcript per Million mapped reads (FPKM). Differentially expressed genes (DEGs) were identified using the R (Version 4.3.2) package DESeq2. The *p*-values obtained were adjusted using the Benjamini–Hochberg method. Genes that met the criteria of |Log2(Fold change)| ≥ 1 and *FDR* < 0.05 were considered as DEGs. TBtools (Version 2.056) was used to extract the protein sequences of the DEGs, which were annotated in the eggNOG Mapper database (http://eggnog-mapper.embl.de/, accessed on 10 June 2024). GO (Gene Ontology) (http://geneontology.org/, accessed on 10 June 2024) and KEGG (http://www.genome.jp/kegg/pathway.html, accessed on 10 June 2024) enrichment analyses were conducted to identify functional annotations related to metabolic pathways. Visualizations were generated using TBtools (Version 2.056).

## 5. Conclusions

In this study, through an in-depth analysis of the genomes of three medicinal *Glycyrrhiza* species, we successfully identified the *PAL* gene family and systematically investigated its fundamental characteristics. Further research revealed that *GuPAL1* plays a crucial role in stress response, plant growth, and flavonoid biosynthesis. Our findings provide a foundation for elucidating the molecular mechanisms of flavonoid metabolism in *Glycyrrhiza* and offer theoretical support for improving *Glycyrrhiza* quality and stress tolerance through genetic engineering.

## Figures and Tables

**Figure 1 ijms-26-04073-f001:**
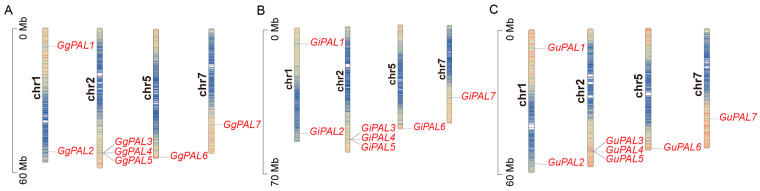
Distribution map of *PAL* family genes on the chromosomes of three *Glycyrrhiza* species. (**A**–**C**) correspond to *G. glabra*, *G. inflata*, and *G. uralensis*, respectively. “chr1”, “chr2”, etc., represent different chromosomes, and different colors indicate different characteristics of the chromosomes. The *PAL* genes are marked in red, and the arrows indicate their positions on the chromosomes. The scale on the left side of the chromosome represents the chromosome length (in Mb).

**Figure 2 ijms-26-04073-f002:**
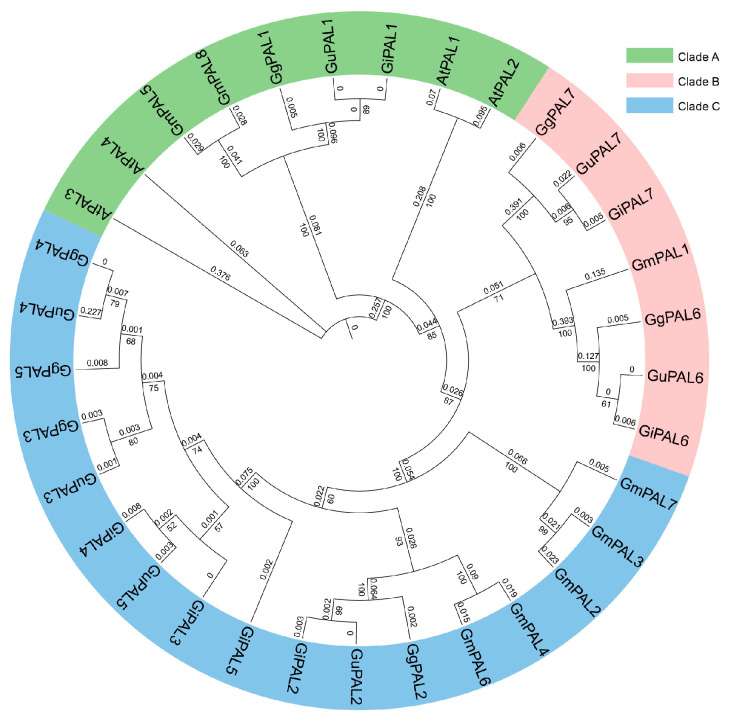
Phylogenetic tree of *PAL* family genes in *G. uralensis*, *G. inflata*, *G. glabra*, *G. max*, and *A. thaliana.* “Gu”, “Gi”, “Gg”, “Gm”, and “At” represent *G. uralensis*, *G. inflata*, *G. glabra*, *G. max*, and *A. thaliana*, respectively. The phylogenetic tree is divided into three branches: Clade A, Clade B, and Clade C. The gene names on the branches represent different *PAL* gene members of each species, and the phylogenetic tree shows the degree of genetic relationship among these gene members.

**Figure 3 ijms-26-04073-f003:**
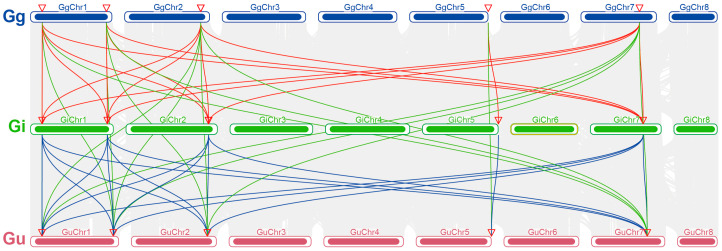
Collinearity relationship diagram of *PAL* family genes among chromosomes of *G. uralensis*, *G. inflata*, and *G. glabra.* Gu, Gi, and Gg represent *G. uralensis*, *G. inflata*, and *G. glabra*, respectively. “GgChr–GgChr8”, “GiChr1–GiChr8”, and “GuChr1–GuChr7” represent different chromosomes of the corresponding *Glycyrrhiza* species. Different colored lines connect the chromosomes of different species, with each line representing a collinear region between chromosomes, indicating the conserved order and orientation of genes in these regions.

**Figure 4 ijms-26-04073-f004:**
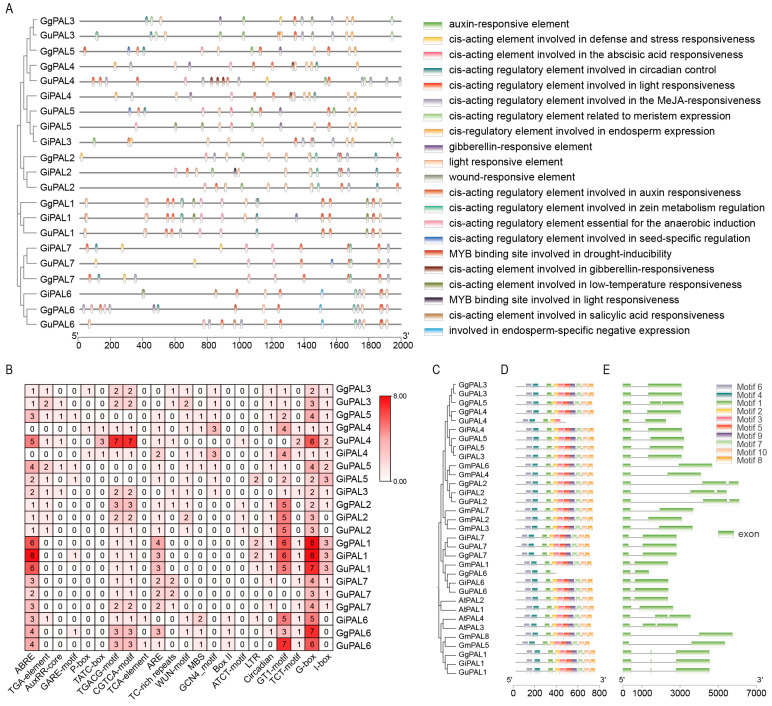
Analysis of promoter *cis*-acting element, conserved motif, and gene structure of *PAL* family genes in *G. uralensis*, *G. inflata*, *G. glabra*, *G. max*, and *A. thaliana*: (**A**) Analysis of the distribution of *cis*-acting elements in the *PAL* promoter region. Different-colored ellipses represent different types of elements. The horizontal axis represents the base positions from the 5′ to the 3′ end of the promoter. (**B**) Presents of the number of different *cis*-acting elements in each *PAL* gene in the form of a heat map, which is reflected by the depth of color. The specific values are marked within the squares. (**C**) Phylogenetic tree-based evolutionary relationship analysis of *PAL* genes from different species. The branches reflect the degree of genetic relatedness. (**D**) Conserved motif analysis of the *PAL* genes from different species. Squares of different colors represent different motifs, visually presenting the types and positions of motifs contained in the genes. (**E**) Gene structure analysis of the *PAL* genes. The black lines and green bars represent the introns and exons, respectively.

**Figure 5 ijms-26-04073-f005:**
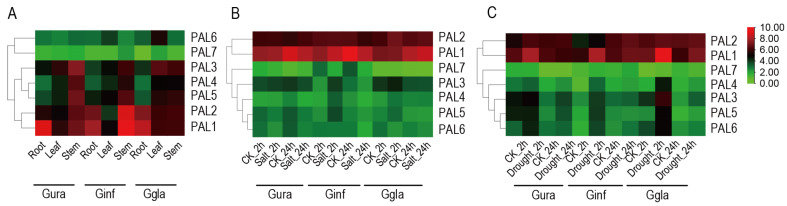
Expression analysis of *PALs* in various tissues and under different treatments in three *Glycyrrhiza* species: (**A**) Expression levels of *PALs* in different tissues of three *Glycyrrhiza* species. The horizontal axis represents different tissue types, including Root, Leaf, and Stem. (**B**) Expression levels of *PAL* genes in the roots of three *Glycyrrhiza* species under 200 mM NaCl treatment. Each numerical value represents the average relative expression level obtained from three independent replicate experiments. (**C**) Expression levels of *PAL* genes in the roots of three *Glycyrrhiza* species under 10% PEG treatment. Each value in the figure represents the average relative expression level obtained from three replicate experiments. Gura, Ggla, and Ginf represent *G. uralensis*, *G. glabra*, and *G. inflata*, respectively.

**Figure 6 ijms-26-04073-f006:**
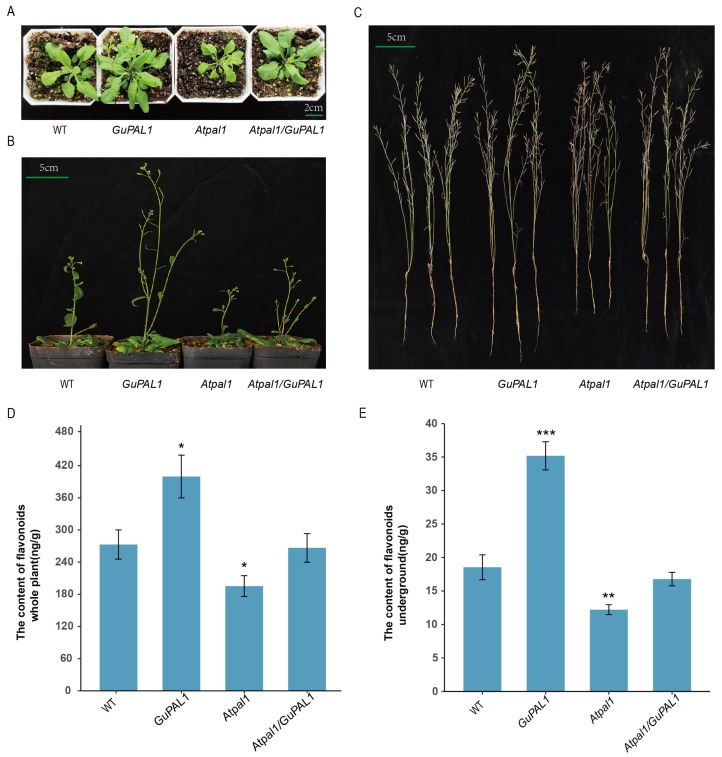
Analysis of phenotypes and flavonoid contents of the transgenic *A. thaliana* lines: (**A**–**C**) are the growth status of 15-, 25-, and 45-day WT plants, overexpressing *GuPAL1* plants, *Atpal1* mutant plants, and *Atpal1* functional complementation plants, respectively. WT: Wild type; *Atpal1*: *Atpal mutant*; *Atpal1*/*GuPAL1*: *Atpal1* functional complementation lines. Bar = 1 cm. (**D**,**E**) represent the bar charts of flavonoid contents in the intact plant and underground parts, respectively, of 45-day-old WT, *GuPAL1*, *Atapl1 mutant*, and Atpal1/*GuPAL1 A. thaliana* plants, using the 45-day-old plant materials as samples. (mean ± standard error, n = 3). The *t*-test was used for significant difference analysis between WT and GuPAL1-OE plants, WT and *Atpal1* mutant, WT and *Atpal1*/*GuPAL1* complementation lines, with *, **, and *** denoting significant differences at *p* < 0.05, 0.01, and 0.001 levels, respectively.

**Figure 7 ijms-26-04073-f007:**
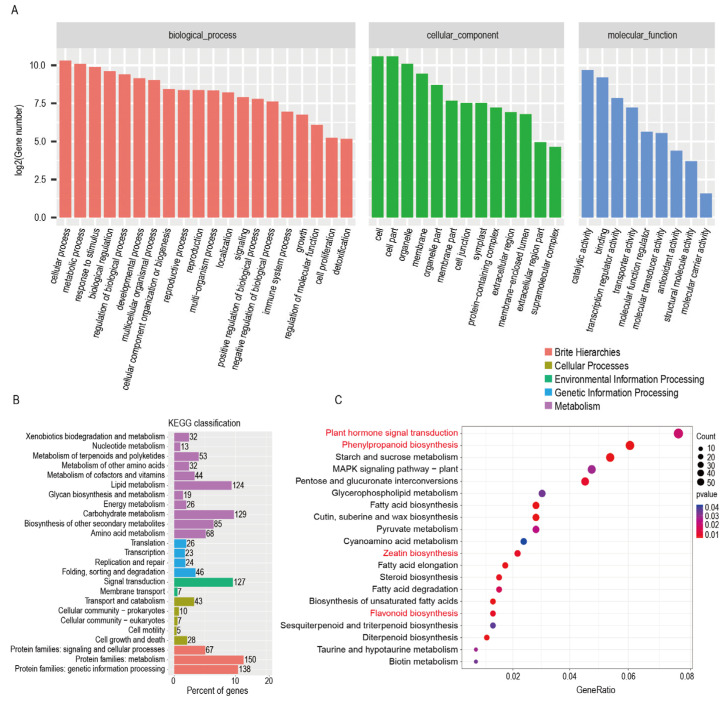
Transcriptomic analysis of *A. thaliana* overexpressing *GuPAL1*: (**A**) GO annotation of up-regulated genes in the transcriptome of *GuPAL1*-OE *A. thaliana.* (**B**) KEGG annotation of up-regulated genes in the transcriptome of *GuPAL1*-OE *A. thaliana.* (**C**) KEGG enrichment of up-regulated genes in the transcriptome of *GuPAL1*-OE *A. thaliana*.

**Figure 8 ijms-26-04073-f008:**
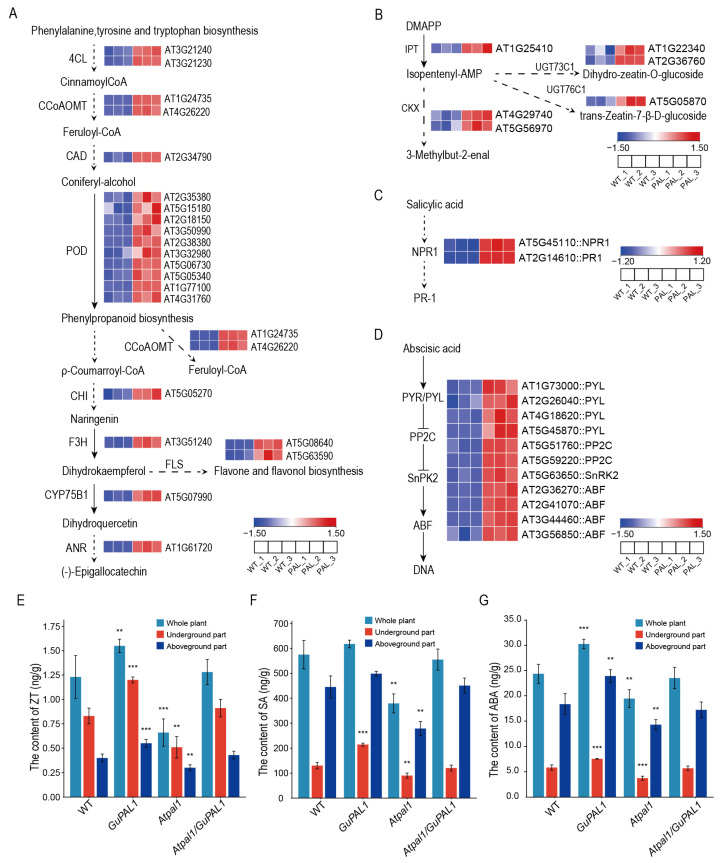
Transcriptomic expression analysis and content detections of pathways of flavonoid, ZT, SA, and ABA in transgenic *Arabidopsis.* Heatmaps of the expressions of up-regulated genes located in the pathways of flavonoid biosynthesis (**A**), ZT biosynthesis (**B**), and SA signaling (**C**,**D**). (**E**–**G**) Detections of the contents of ZT (**E**), SA (**F**), and ABA (**G**) in different parts of the transgenic *Arabidopsis.* The samples from 45-day-old *A. thaliana* plants of WT, *GuPAL1*, *Atpal1* mutant, and *Atpal1*/*GuPAL1* were collected for analysis. The *t*-test was used for significant difference analysis between WT and *GuPAL1*-OE plants, WT and *Atpal1* mutant, WT and *Atpal1*/*GuPAL1* complementation lines, with **, and *** denoting significant differences at *p* < 0.01 and 0.001 levels, respectively.

**Figure 9 ijms-26-04073-f009:**
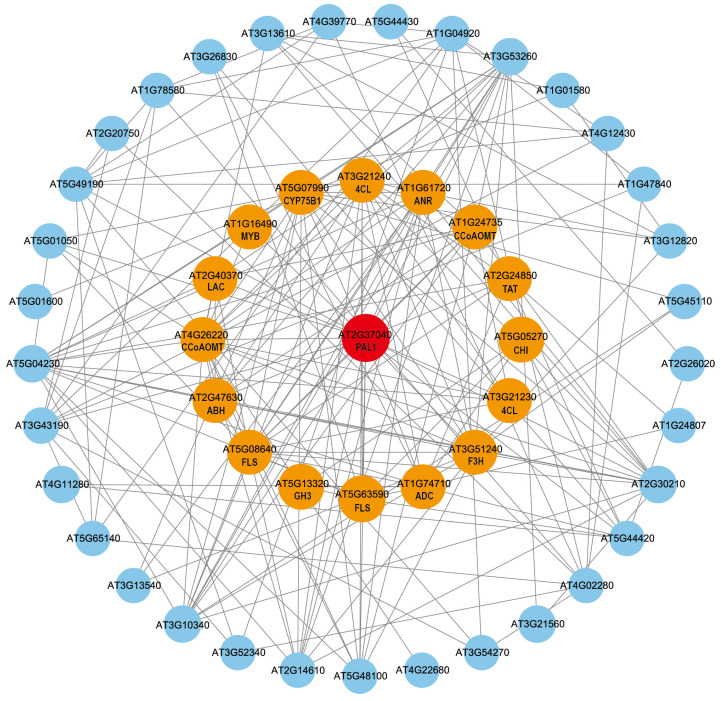
Protein–Protein Interaction (PPI) Analysis of *GuPAL1*. Red, orange, and blue colors represent *GuPAL1*, PPI-predicted genes that directly interact with *GuPAL1*, and other flavonoid and plant hormone-related pathway differentially co-expressed genes (*DCGs*), respectively. Hierarchical clustering analysis was performed using Euclidean distance and the complete linkage method.

## Data Availability

All data are presented in the article and the Appendix A.

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
