# Peer review of "Overexpression of the Glycyrrhiza uralensis Phenylalanine Ammonia-Lyase Gene GuPAL1 Promotes Flavonoid Accumulation in Arabidopsis thaliana"

_ijms, 2025, doi:10.3390/ijms26094073_

Round 1
Reviewer 1 Report
Comments and Suggestions for Authors
The manuscript presents a comprehensive investigation of the phenylalanine ammonia-lyase (PAL) gene family members in three Glycyrrhiza species. Based on the gene expression analysis of 7 PALs in different tissues and after stress treatments, the candidate gene GuPAL1 was identified. The function of GuPAL1, which is involved in flavonoid biosynthesis and related to hormone accumulation, was further verified by constructing GuPAL1-OE or atpal1/GuPAL-OE transgenic Arabidopsis thaliana. The study is well-organized, and the conclusions are reliable. However, several issues should be addressed before acceptance.
- In the phylogenetic tree shown in Figure 2, please provide the bootstrap values of the branches and the evolutionary distances between branches.
- Provide more details on the number of biological and technical replicates for RT-qPCR of 7 PALs, RNA-seq, and hormone quantification in Methods and the Figure legends.
- Please supplement the DNA identification and GuPAL1 expression profiles of transgenic Arabidopsis thaliana.
- Add scale bars in Figures 6A-C.
- In Figures 6D-E, please change the method of statistics analysis. The Dunnett test following one-way ANOVA will be more appropriate.
- Please provide more details on the statistics and significance analysis of the data in Figures 8E-G.
Author Response
Reviewer 1
The manuscript presents a comprehensive investigation of the phenylalanine ammonia-lyase (PAL) gene family members in three Glycyrrhiza species. Based on the gene expression analysis of 7 PALs in different tissues and after stress treatments, the candidate gene GuPAL1 was identified. The function of GuPAL1, which is involved in flavonoid biosynthesis and related to hormone accumulation, was further verified by constructing GuPAL1-OE or atpal1/GuPAL-OE transgenic Arabidopsis thaliana. The study is well-organized, and the conclusions are reliable. However, several issues should be addressed before acceptance.
- In the phylogenetic tree shown in Figure 2, please provide the bootstrap values of the branches and the evolutionary distances between branches.
Response: Thank you for the advice. As requested, we have now updated Figure 2 to include bootstrap values (shown at branch nodes) and evolutionary distances (indicated on the scale bar).
- Provide more details on the number of biological and technical replicates for RT-qPCR of 7 PALs, RNA-seq, and hormone quantification in Methods and the Figure legends.
Response: Thank you for the advice. We have performed three biological replicates for each sample, and we have modified the descriptions in the revised manuscript.
3、Please supplement the DNA identification and GuPAL1 expression profiles of transgenic Arabidopsis thaliana.
4、Add scale bars in Figures 6A-C.
Response: We thank the reviewer for this helpful suggestion. We have now added clearly visible scale bars to each panel of Figure 6 (Figures 6A-C) as requested.
- In Figures 6D-E, please change the method of statistics analysis. The Dunnett test following one-way ANOVA will be more appropriate.
Response: We appreciate the reviewer's suggestion regarding statistical analysis. In our study, we specifically aimed to compare each transgenic line (GuPAL1-OE plants, Atpal1 mutant, and Atpal1/GuPAL1 complementation lines) directly with the WT control, rather than performing multiple comparisons among all groups.This approach is clearly stated in the figure legend of Figure 6.
6、Please provide more details on the statistics and significance analysis of the data in Figures 8E-G.
Response: We appreciate the reviewer's request for clarification regarding the statistical analysis in Figures 8E-G. Similar to the approach used in Figures 6D-E.

Reviewer 2 Report
Comments and Suggestions for Authors
Overall Suggestions
The manuscript entitled “Overexpression of the Glycyrrhiza uralensis phenylalanine ammonia-lyase gene GuPAL1 promotes the flavonoids accumulation in Arabidopsis” explores the identification of the PAL gene family in three medicinal Glycyrrhiza species and investigates the specific function of GuPAL1 in promoting plant growth and flavonoid biosynthesis when overexpressed in Arabidopsis thaliana.The manuscript’s English is overall acceptable; however, there are a few long or complex sentences that could be simplified for clarity. Some abbreviations (e.g., “OE,” “ABA,” “WT,” “SA”) should be consistently introduced/explained at first mention and used uniformly throughout. The study is well structured, presenting both in silico analyses (phylogeny, promoter cis-elements, and synteny) and in vivo experiments (phenotypic evaluation, flavonoid measurements, and transcriptome profiling). Overall, the work is scientifically sound and addresses an important biological question regarding the regulation of secondary metabolism in a valuable medicinal plant genus. The experiments on Arabidopsis (OE, mutant, and complementation lines) convincingly support the role of GuPAL1 in flavonoid accumulation and plant growth. The transcriptome analysis nicely identifies the upregulated pathways (phenylpropanoid, flavonoid biosynthesis, plant hormone signaling), indicating a comprehensive approach. The study would benefit from clearer emphasis on how these findings can be leveraged for Glycyrrhiza cultivation or breeding, in other words, practical application. The manuscript is well organized and addresses a relevant topic in plant secondary metabolism and functional genomics. The main experiments are sufficiently described, though a bit more elaboration on the practical implications for Glycyrrhiza breeding would be helpful. Overall, the text would benefit from some linguistic refinements, the only major overhauls necessary, in my opinion, is to remake figures 4 and 7, due to their information being virtually unreadble with major zooming and loss of quality.
Specific Suggestions
Title
“Overexpression of the Glycyrrhiza uralensis phenylalanine ammonia-lyase gene GuPAL1 pro-
motes the flavonoids accumulation in Arabidopsis” It would be better to change the title to “…promotes flavonoid accumulation in Arabidopsis thaliana.” for smoother reading.
L77-79
“This study aims to explore the relevant molecular mechanisms, provide a theoretical basis for understanding the effects of salt and drought stress on flavonoid accumulation, and offer new insights for the high-quality cultivation and quality improvement of G. uralensis.” Break this into two sentences or reduce length
Figure 4 has hard to read labels, they are to crowded with information and are very hard to read. The same can be stated for figure 7.
L285-287
“Further analysis indicated significant enrichment in pathways such as plant hormone signal transduction, flavonoid biosynthesis, phenylpropanoid biosynthesis…” Add one sentence on why hormone signaling might be relevant for GuPAL1 function
L307-310
“Similarly, in the flavonoid metabolic pathway, the up-regulated genes included those encoding chalcone isomerase (CHI), flavanone 3—hydroxylase (F3H), flavonol synthase (FLS), cytochrome P450 monooxygenase 75B1 (CYP75B1), and anthocyanidin reductase (ANR).” Provide a brief statement about how this set of enzymes connects to the observed increase in flavonoids. The authors can elaborate better on the link between gene expression data and phenotypic results.
L349-350
“…These genes are pivotal for flavonoid biosynthesis in Glycyrrhiza, laying a solid foundation for subsequent integrated metabolomic analysis…” Clarify how these interactions can be exploited or studied further
L522-525
“…This discovery reveals the multifaceted role of GuPAL1 in plant metabolic regulation and provides new strategies for enhancing flavonoid content and optimizing hormone synthesis in plants, while also offering a new perspective for cross-pathway metabolic engineering research.” Conclude with a concise statement explicitly referring to the potential applications in Glycyrrhiza cultivation and breeding
Author Response
Reviewer 2
Overall Suggestions
The manuscript entitled “Overexpression of the Glycyrrhiza uralensis phenylalanine ammonia-lyase gene GuPAL1 promotes the flavonoids accumulation in Arabidopsis” explores the identification of the PAL gene family in three medicinal Glycyrrhiza species and investigates the specific function of GuPAL1 in promoting plant growth and flavonoid biosynthesis when overexpressed in Arabidopsis thaliana.The manuscript’s English is overall acceptable; however, there are a few long or complex sentences that could be simplified for clarity. Some abbreviations (e.g., “OE,” “ABA,” “WT,” “SA”) should be consistently introduced/explained at first mention and used uniformly throughout.
Response: We sincerely appreciate the reviewer’s careful attention to detail. We have now ensured that all abbreviations are explicitly defined upon their first appearance in the main text, figure legends, and supplementary materials.
The study is well structured, presenting both in silico analyses (phylogeny, promoter cis-elements, and synteny) and in vivo experiments (phenotypic evaluation, flavonoid measurements, and transcriptome profiling). Overall, the work is scientifically sound and addresses an important biological question regarding the regulation of secondary metabolism in a valuable medicinal plant genus. The experiments on Arabidopsis (OE, mutant, and complementation lines) convincingly support the role of GuPAL1 in flavonoid accumulation and plant growth. The transcriptome analysis nicely identifies the upregulated pathways (phenylpropanoid, flavonoid biosynthesis, plant hormone signaling), indicating a comprehensive approach. The study would benefit from clearer emphasis on how these findings can be leveraged for Glycyrrhiza cultivation or breeding, in other words, practical application. The manuscript is well organized and addresses a relevant topic in plant secondary metabolism and functional genomics. The main experiments are sufficiently described, though a bit more elaboration on the practical implications for Glycyrrhiza breeding would be helpful. Overall, the text would benefit from some linguistic refinements, the only major overhauls necessary, in my opinion, is to remake figures 4 and 7, due to their information being virtually unreadble with major zooming and loss of quality.
Specific Suggestions
Title
“Overexpression of the Glycyrrhiza uralensis phenylalanine ammonia-lyase gene GuPAL1 pro-motes the flavonoids accumulation in Arabidopsis” It would be better to change the title to “…promotes flavonoid accumulation in Arabidopsis thaliana.” for smoother reading.
Response: We sincerely appreciate the reviewer's suggestion. We have revised the title as recommended to: Overexpression of the Glycyrrhiza uralensis phenylalanine ammonia-lyase gene GuPAL1 promotes flavonoid accumulation in Arabidopsis thaliana.
L77-79
“This study aims to explore the relevant molecular mechanisms, provide a theoretical basis for understanding the effects of salt and drought stress on flavonoid accumulation, and offer new insights for the high-quality cultivation and quality improvement of G. uralensis.” Break this into two sentences or reduce length
Response: We have revised the statement as suggested. The original sentence has been split into two clearer statements, see L79-L83.
Figure 4 has hard to read labels, they are to crowded with information and are very hard to read. The same can be stated for figure 7.
Response: We sincerely appreciate this constructive feedback. We have carefully revised both figures to improve readability.
L285-287
“Further analysis indicated significant enrichment in pathways such as plant hormone signal transduction, flavonoid biosynthesis, phenylpropanoid biosynthesis…” Add one sentence on why hormone signaling might be relevant for GuPAL1 function
Response: We appreciate this valuable suggestion. As requested, we have added the following explanation in Lines 310-312.
L307-310
“Similarly, in the flavonoid metabolic pathway, the up-regulated genes included those encoding chalcone isomerase (CHI), flavanone 3—hydroxylase (F3H), flavonol synthase (FLS), cytochrome P450 monooxygenase 75B1 (CYP75B1), and anthocyanidin reductase (ANR).” Provide a brief statement about how this set of enzymes connects to the observed increase in flavonoids. The authors can elaborate better on the link between gene expression data and phenotypic results.
Response: We thank the reviewer for this constructive suggestion. As requested, we have added the following clarification in Lines 335-339: This coordinated upregulation suggests a metabolic flux redirection towards flavonoid production, with CHI and F3H enhancing the core flavonoid biosynthesis, FLS and CYP75B1 promoting flavonol diversification, and ANR facilitating the conversion to proanthocyanidins, collectively contributing to the observed increase in total flavonoid content in GuPAL1-OE plants.
L349-350
“…These genes are pivotal for flavonoid biosynthesis in Glycyrrhiza, laying a solid foundation for subsequent integrated metabolomic analysis…” Clarify how these interactions can be exploited or studied further
Response: We appreciate this insightful suggestion. As requested, we have added the following clarification in Lines 385-389:These interacting genes provide crucial targets for subsequent metabolomic analyses, enabling in-depth elucidation of flavonoid biosynthetic mechanisms through approaches like CRISPR-based validation of key regulatory nodes and construction of regulatory networks.
L522-525
“…This discovery reveals the multifaceted role of GuPAL1 in plant metabolic regulation and provides new strategies for enhancing flavonoid content and optimizing hormone synthesis in plants, while also offering a new perspective for cross-pathway metabolic engineering research.” Conclude with a concise statement explicitly referring to the potential applications in Glycyrrhiza cultivation and breeding
Response: We thank the reviewer for this valuable suggestion. As requested, we have added the following application-oriented conclusion in Lines 357-360: These findings position GuPAL1 as a prime molecular target for genetically improving Glycyrrhiza varieties, enabling simultaneous enhancement of medicinal flavonoid production and stress adaptability in cultivation practices.

Reviewer 3 Report
Comments and Suggestions for Authors
The manuscript presents results from an investigation on licorice PAL genes in Arabidopsis. It is interesting to see how the family of the pal genes express differently with respect to different plant organelles, and that they also affect expression of the other genes of the flavonoid biosynthesis pathway. I would like to see more of these connections explained as background information in the introduction part.
With respect to the introduction, lines 102-107 list results that obviously should be under the Result-section and not in the introduction part.
The results section uses much space for phylogenetic analysis (section 2.3) and collinearity relationship (2.4) which I find less interesting as these are not primary results. I would recommend to place the most breaking news first. As highlighted in the abstract transcriptomic analysis demonstrated that several key genes in the phenylpropanoid and flavonoid pathways (4CL, CCoAOMT, CAD, POD, F3H, FLS) were significantly enriched. How is this connected to the expression of PAL?
Include a reference to the seed treatment with conc sulphuric acid (line 444).
Paragraph 4.5 doesn’t provide enough information to verify the findings with respect to analysis of flavonoids. Sum of flavonoids? Which flavonoids? How where they determined? Are they licorice or Arabidopsis flavonoids?
Author Response
Reviewer 3
The manuscript presents results from an investigation on licorice PAL genes in Arabidopsis. It is interesting to see how the family of the pal genes express differently with respect to different plant organelles, and that they also affect expression of the other genes of the flavonoid biosynthesis pathway. I would like to see more of these connections explained as background information in the introduction part.
Response: We express our heartfelt gratitude to the reviewer for the invaluable comments. Concerning the interrelationships among the expression of the PAL gene family, and the regulation of the flavonoid biosynthetic pathway, we have already elaborated on these aspects in the initial Introduction section (Lines 86-100). Specifically, our discussion encompasses the following: ...The central and indispensable role of phenylalanine ammonia-lyase (PAL) as a pivotal rate-limiting enzyme that serves as a bridge between primary and secondary metabolism in the metabolic pathway [44-45]. The regulatory mechanisms through which the activity and expression of PAL influence the downstream biosynthesis of flavonoids [46-47]. The broader and more intricate regulatory network governing the modulation of PAL by diverse environmental cues and hormonal factors [24-42]... If the reviewer believes that additional elaboration or modification is necessary, we are fully committed to making comprehensive enhancements and refinements during the subsequent revision phase.
With respect to the introduction, lines 102-107 list results that obviously should be under the Result-section and not in the introduction part.
Response: We sincerely appreciate the reviewer's careful reading and constructive suggestion regarding the organization of our introduction. While we agree that detailed results should be presented in the Results section, we intentionally included a brief overview of key findings (Lines 107-113) in the introduction to Provide readers with a clear roadmap of the study's major discoveries. That said, we have now refined this section.
The results section uses much space for phylogenetic analysis (section 2.3) and collinearity relationship (2.4) which I find less interesting as these are not primary results. I would recommend to place the most breaking news first. As highlighted in the abstract transcriptomic analysis demonstrated that several key genes in the phenylpropanoid and flavonoid pathways (4CL, CCoAOMT, CAD, POD, F3H, FLS) were significantly enriched. How is this connected to the expression of PAL?
Response: We thank the reviewer for this constructive suggestion. As clearly described in our manuscript (Lines 86-100, 111-113), PAL functions as the upstream regulator of both phenylpropanoid and flavonoid biosynthesis pathways (including 4CL, CCoAOMT, CAD, POD, F3H, and FLS). Our experimental results demonstrate that PAL gene overexpression in Arabidopsis thaliana effectively modulates the expression of these downstream genes in the metabolic pathways.
Include a reference to the seed treatment with conc sulphuric acid (line 444).
Response: Thank you for your suggestion. We have added the reference to seed treatment with concentrated sulphuric acid (Line 493) in the Methods section.
Paragraph 4.5 doesn’t provide enough information to verify the findings with respect to analysis of flavonoids. Sum of flavonoids? Which flavonoids? How where they determined? Are they licorice or Arabidopsis flavonoids?
Response: Thank you for your suggestion. We have supplemented the Method 4.5 section with detailed information about the standard substances used for measuring flavonoids and hormones (including IAA, ZT, BR, SA, ABA, MeJA, GA3, and Rutin). The standard curves for each compound are provided in Supplementary Table S3. Specifically, we quantified Rutin as a representative flavonoid in T3 generation transgenic Arabidopsis using LC-MS/MS analysis.
